# Quantitative magnetic resonance imaging predicts individual future liver performance after liver resection for cancer

Damian J. Mole[1,2], Jonathan A. Fallowfield[2], Ahmed E. Sherif[1,3], Timothy Kendall[4,5], Scott Semple[6], Matt Kelly[7], Gerard Ridgway[7], John J. Connell[7]*, John McGonigle[7], Rajarshi Banerjee[7], J. Michael Brady[7], Xiaozhong Zheng[2], Michael Hughes[1], Lucile Neyton[2], Joanne McClintock[8], Garry Tucker[9], Hilary Nailon[9], Dilip Patel[10], Anthony Wackett[11], Michelle Steven[11], Fenella Welsh[8], Myrddin Rees[8], the HepaT1ca Study Group[¶]

1 Clinical Surgery, Royal Infirmary of Edinburgh, University of Edinburgh, Edinburgh, United Kingdom, 2 Centre for Inflammation Research, University of Edinburgh, Queen's Medical Research Institute, Edinburgh, United Kingdom, 3 Department of HPB Surgery, National Liver Institute, Menoufia University, Shibin Elkom, Egypt, 4 Institute of Genetics and Molecular Medicine, Edinburgh, United Kingdom, 5 Department of Pathology, NHS Lothian, Edinburgh, United Kingdom, 6 Centre for Cardiovascular Science, Queen's Medical Research Institute, University of Edinburgh, Edinburgh, United Kingdom, 7 Perspectum, Gemini One, Oxford, United Kingdom, 8 Hampshire Hospitals Foundation Trust, Basingstoke, United Kingdom, 9 Clinical Research Facility, NHS Lothian, Edinburgh, United Kingdom, 10 Clinical Radiology, NHS Lothian, Edinburgh, United Kingdom, 11 Edinburgh Clinical Trials Unit, Edinburgh, United Kingdom

¶ Membership of the HepaT1ca Study Group is provided in the Acknowledgments.
* john.connell@perspectum.com

**Data Availability Statement:** All relevant data are provided at https://datashare.is.ed.ac.uk/handle/10283/3716.

## Abstract

The risk of poor post-operative outcome and the benefits of surgical resection as a curative therapy require careful assessment by the clinical care team for patients with primary and secondary liver cancer. Advances in surgical techniques have improved patient outcomes but identifying which individual patients are at greatest risk of poor post-operative liver performance remains a challenge. Here we report results from a multicentre observational clinical trial (ClinicalTrials.gov NCT03213314) which aimed to inform personalised pre-operative risk assessment in liver cancer surgery by evaluating liver health using quantitative multiparametric magnetic resonance imaging (MRI). We combined estimation of future liver remnant (FLR) volume with corrected T1 (cT1) of the liver parenchyma as a representation of liver health in 143 patients prior to treatment. Patients with an elevated preoperative liver cT1, indicative of fibroinflammation, had a longer post-operative hospital stay compared to those with a cT1 within the normal range (6.5 vs 5 days; p = 0.0053). A composite score combining FLR and cT1 predicted poor liver performance in the 5 days immediately following surgery (AUROC = 0.78). Furthermore, this composite score correlated with the regenerative performance of the liver in the 3 months following resection. This study highlights the utility of quantitative MRI for identifying patients at increased risk of poor post-operative liver performance and a longer stay in hospital. This approach has the potential to inform the assessment of individualised patient risk as part of the clinical decision-making process for liver cancer surgery.

**Funding:** We gratefully acknowledge the funders of this work through an Early Stage Biomedical Catalyst Award from Innovate UK, part of UK Research and Innovation, ref: 79891-510341 to MK. DJM is funded by the UK Medical Research Council through a Senior Clinical Fellowship, ref: MR/P008887/1. JAF was funded through an NHS Research Scotland/Universities Senior Clinical Fellowship. Additional funding was contributed by North Hampshire Medical Fund. The funder provided support in the form of salaries for authors, but did not have any additional role in the study design, data collection and analysis, decision to publish, or preparation of the manuscript. The specific roles of these authors are articulated in the 'author contributions' section. MK, GR, JJC, JMcG, RB, and JMB are employed by Perspectum Ltd.

**Competing interests:** I have read the journal's policy and the authors of this manuscript have the following competing interests: MK, GR, JJC, JMcG, RB, and JMB are employees and shareholders at Perspectum Ltd. This does not alter our adherence to PLOS ONE policies on sharing data and materials. There are no other actual or perceived conflicts of interest to declare.

## Introduction

Liver resection for primary and secondary malignant tumours is a fundamental component of the multimodal treatment of cancer [1, 2]. When an individual patient and multidisciplinary surgical team is considering liver resection, a major challenge is knowing in advance whether liver surgery will be survivable and how that individual's liver will recover function after surgery [3]. We term this future liver performance (FLP), and misjudging FLP can result in post-surgical liver failure and death [4].

Future liver performance is a function of future liver remnant (FLR) volume, FLR function and liver regenerative capacity. FLR volume is typically straightforward to calculate from routine pre-operative imaging with computed tomography (CT) or magnetic resonance imaging (MRI) [5]. However, this requires substantial time input from specialised radiologists [5] using dedicated software tools to accurately delineate the planned FLR.

Estimates of FLR function are based on clinical risk factors, with older age or presence of liver disease, obesity [6] or diabetes mellitus/metabolic syndrome [7] all indicative of poor FLP [8]. When combined with laboratory indices of liver synthetic function (e.g., serum albumin concentration or prothrombin time), these risk factors give a reasonable estimate of FLP to an experienced liver surgeon [8]. More objective direct estimates of hepatocellular function can be obtained using $^{99m}$Tc-mebrofenin scintigraphy [9], $^{13}$C-methacetin metabolism [10], or indocyanine green (ICG) clearance [11]. However, these methods require tracer injection and specialised detection devices and have not been widely adopted by most centres offering liver resection, with the exception of ICG clearance in Asia [12]. Various blood-based composite risk scores have been created to estimate survival after liver resection including the 50–50 criteria of prothrombin time and bilirubin [13], peak bilirubin >7mg/dL [14] and the Hyder-Pawlik [4] composite score of creatinine, bilirubin, INR and the Clavien-Dindo grade; these scores are all calculated on measurements taken in the time recovering after surgery and are less sensitive to mild morbidities where changes in clinical management may still be required.

Poor liver regenerative capacity is difficult to predict accurately and to detect pre-operatively when interventions intended to increase the FLR volume (FLRV) such as portal vein embolization [15] or combined portal and hepatic vein occlusion, fail to induce hypertrophy [16]. For an otherwise normal healthy liver affected by cancer, a FLRV of at least 5 cm$^3$ per kg body weight, or 25% of pre-operative size is taken as a safe threshold below which an individual undergoing liver resection is generally considered to have an unacceptable risk of post-operative liver failure. For livers with evidence of parenchymal disease (e.g., cirrhosis) a higher threshold 40% is commonly used [17]. These thresholds have been derived from cohort studies and are based on cumulative experience of patients undergoing liver resection and do not take into account individual variations in liver health. Patients with very healthy livers could tolerate smaller FLRVs and conversely, post-operative liver failure can occur in patients with FLRV above the usual thresholds, often unexpectedly. Furthermore, some patients receiving systemic anti-cancer therapy may also develop hepatica injury, including chemotherapy-associated steatohepatitis, further increasing the risks associated with liver resection [18]. There is therefore a great and unmet need to develop a non-invasive imaging technology that can accurately and reliably quantify FLP when used pre-operatively.

Liver*MultiScan* is a non-invasive quantitative multiparametric MRI (mpMRI) technology [19] that has shown high diagnostic accuracy for liver fibroinflammation [20], steatosis and haemosiderosis [21], as well as predicting liver-related outcomes [22] in patients with chronic liver disease. It has also been demonstrated to be cost-effective in stratifying patients with non-alcoholic fatty liver disease [23] and sensitive to changes in non-alcoholic steatohepatitis [24].

The aim of this study was to evaluate, for the first time, the utility of combining these quantitative mpMRI metrics of pre-operative liver health with estimated FLR to forecast individualised risk of poor liver performance in patients undergoing surgical resection for cancer in the liver.

## Methods

This study was conducted in accordance with the principles of the Declaration of Helsinki 2013 and approved by the institutional research departments, the South East Scotland Research Ethics Committee 02 (Reference: 17/SS/0049), NHS Scotland R&D and NHS England HRA. The study was registered prospectively on www.clinicaltrials.gov (NCT03213314) and the study design published [25]. Patients being considered for liver resection for cancer were recruited between 7th September 2017 and 31st January 2019 at The Royal Infirmary of Edinburgh or Hampshire Hospitals NHS Foundation Trust with the last patient to enter the study completing the protocol on 31st March 2019. There was no change in treatment, intervention or randomisation in the study. Study data were collected and managed using REDCap electronic data capture tools hosted at the Edinburgh Clinical Trials Unit [26, 27].

### Study assessments

In this observational clinical trial, patients were recruited following clinical diagnosis and treatment planning. 419 individuals were identified at the multidisciplinary cancer meeting and screened, 305 people were approached to take part, of whom 149 participated in the study requiring a pre-operative and a 3-month post-operative study visit at an imaging centre. Relevant clinical data were obtained by trained members of the Clinical Research Facility at each site. Peripheral venous blood samples were obtained from each study participant on each visit and at the time of surgery for multiple routine laboratory analyses and specialist assays. MR images were acquired a mean of 3 days before surgery and 102 days following surgery at either the Candover Clinic Hampshire Hospitals NHS Foundation Trust on a Siemens Aera 1.5T or at the Queen's Medical Research Institute, The University of Edinburgh on a Siemens Skyra 3T MRI scanner. Appropriate 3-plane localiser images were acquired of the abdomen and mpMRI data were collected using non-contrast enhanced multi-slice acquisition protocols along with a 3D image of the liver, totalling approximately 18 minutes.

Clinical care teams were blinded to research MRI scan reports and surgical procedures were performed as standard of care with operative information collected as described in the eCRF including details of the performed operation. Tissue samples were taken from liver resection specimens with strict care being taken not to compromise the direct clinical care pathology reporting. Fresh Tru-cut biopsies of the tumour and a 1 cm$^3$ wedge biopsy of the non-lesion liver parenchyma were snap frozen in liquid nitrogen in the operating theatre immediately after the resection specimen was removed from the patient. Formalin-fixed paraffin-embedded blocks of background liver and tumour tissue were obtained specifically for the study at the time of specimen processing for direct clinical care pathology reporting using the NAFLD activity score (NAS) [28], a summation of the semi-quantitative histological scores for liver fat, inflammation and ballooning. The ISHAK score was used to assess fibrosis [29].

### Image analysis

The Liver volume was delineated from three-dimensional T1-weighted VIBE images using a semi-automatic approach, comprising an automatic deep learning based initialisation followed by manual editing. The deep learning approach used a 3D U-net architecture as previously

described [30]. Six cases were omitted as more than 5% of the liver volume was outside of the field of view of the image acquisition. Tumour locations were defined and guided by the surgical notes and were manually delineated using ITK-SNAP [31] by tracing the tumour in three orthogonal planes and interpolating into a 3D consolidated segmentation. Finally, the delineated liver volume was subdivided into the anatomical Couinaud segments [32], based on manually-placed 3D landmarks (inferior/superior vena cava, right/middle/left hepatic veins, gall bladder fossa, umbilical fissure, left/right portal vein) following the guidelines of Germain et al [33], to facilitate modelling of the planned segmentectomy planes. For atypical wedge resections, a conical frustrum was modelled based on the tumour delineation and the resection path from the liver surface. Surgical notes from the operation plan were used to inform the estimation of FLR volume by removing anatomical segments and wedges as calculated above (Fig 1).

Multislice quantitative cT1, proton density fat fraction (PDFF) and T2* maps were generated using proprietary Liver*MultiScan* software (Perspectum, UK) as previously described [21] with operators blinded to patient status. The quantitative MRI maps were overlaid onto the Couinaud segmented volumetric images to allow characterisation of liver tissue in the FLR.

## Morbidity outcome

Post-operative mortality has previously been modelled by Hyder and Pawlik [4] by combining three weighted blood scores (INR, creatinine, bilirubin) with the Clavien-Dindo score on the third day following surgery, which predicted 90 day mortality following resection for liver cancer. In our study, excellent long-term post-operative patient outcomes were achieved and only five patients exhibited a Hyder-Pawlik score >9.0 (S1 Fig) indicating that this study was underpowered to reach the primary endpoint. However, measures of liver dysfunction in the immediate days following surgery from blood-derived biomarkers (INR, creatinine, bilirubin with Hyder-Pawlik weightings) can serve as an alternative measure of morbidity. We measured and summed these blood-derived biomarkers for each of the five days following surgery and devised the modified Hyder-Pawlik score, to evaluate the risk of morbidity directly after the operation.

## Statistical analysis

Interim analyses were planned in advance. Any errors in data entry were reviewed and corrected with the Chief Investigator's permission. When longitudinal blood sampling was incomplete for the anticipated 5 days (owing to early discharge from hospital), bilirubin, creatinine or INR value was imputed from the value of the previous day. The Hyder-Pawlik score was calculated for each patient based on the values at day 3 post-surgery of bilirubin (mg/dL), creatinine (mg/dL), INR and maximum Clavien-Dindo grade, weighted as previously described [4]. The composite biomarker was generated after a stepwise logistic regression using a modified Hyder-Pawlik score of over 22 as a discriminator and the optimal weighed predictors were selected by Akaike's information criterion. The Youden index identified the optimal receiver operating characteristic.

## Results

Of the 149 patients enrolled in the study, 135 patients underwent liver resection, 7 had portal vein embolization to induce hypertrophy of the FLR and two individuals had transarterial chemoembolisation (Fig 2a). A range of surgical procedures were planned (Fig 2b) resulting in a median [IQR] FLR of 83% [23%–95%]. Demographics and clinicopathologic data are

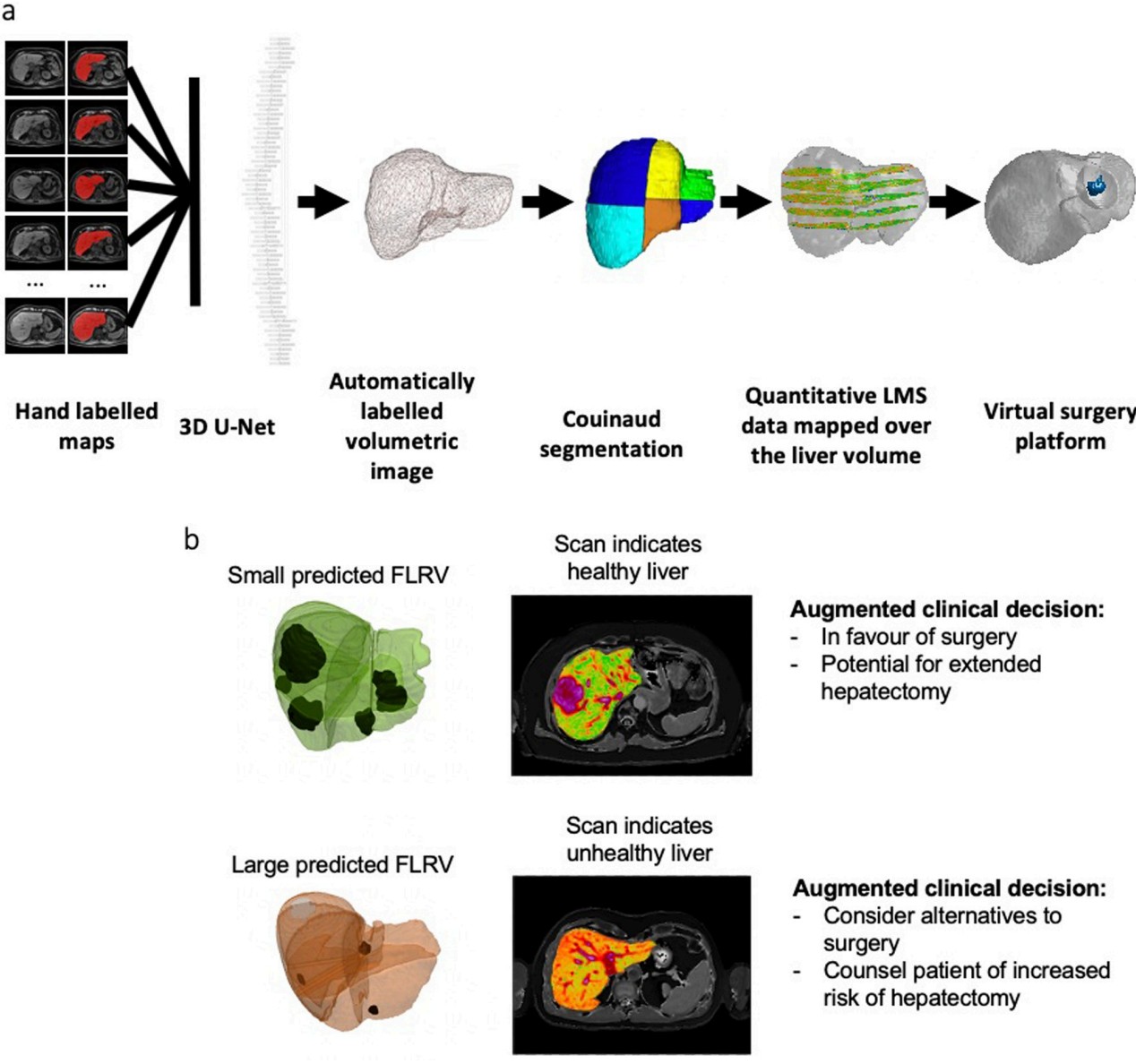

**Fig 1.** (a) Image analysis pipeline. A 3D U-net was trained using hand labelled maps to delineate the liver on 3D T1-weighted images. The surgical plans and anatomical segments of Couinaud are semi-automatically delineated by placing anatomical landmarks. Quantitative tissue characteristics maps are overlaid onto the 3D liver rendering. Segmentectomies or wedge resections defined by the surgical plan are virtually performed to calculate the future liver remnant. (b) Two case studies indicating the proposed utility in clinical decision making.

described in Table 1. The large majority (n = 114, 84% of participants) had liver metastases from colorectal cancer; the remainder had hepatocellular carcinoma (n = 6), cholangiocarci-noma (n = 1) or other metastases (n = 14) including breast cancer metastasis and ovarian can-cer metastasis. 73 patients had received systemic anticancer chemotherapy that was ended a median of 56 days prior to surgery. 12 patients presented with a diagnosed underlying liver disease. MpMRI was performed in all patients and 21% of individuals (29/135) had pre-opera-tive liver cT1 values greater than the upper limit of normal (795 ms) defined for the general population [34, 35].

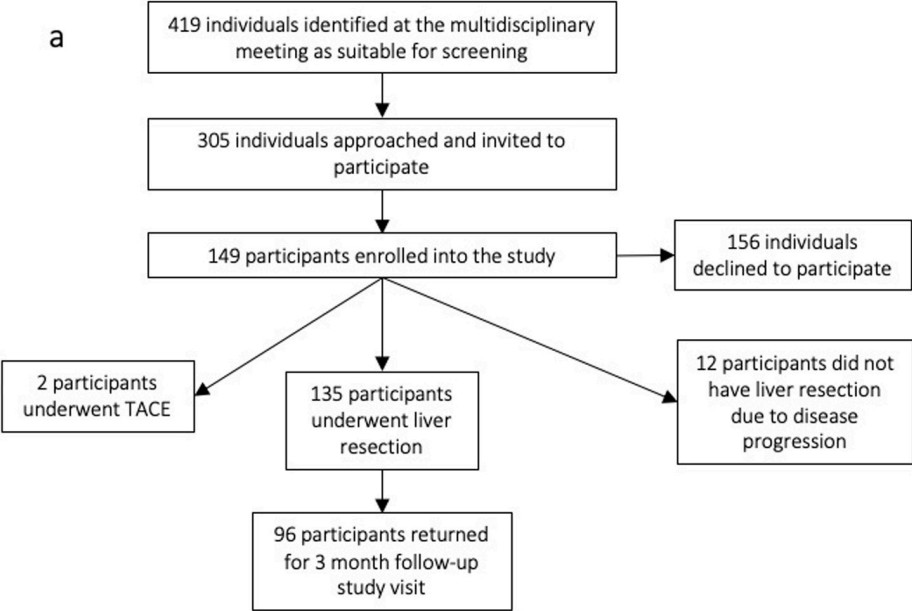

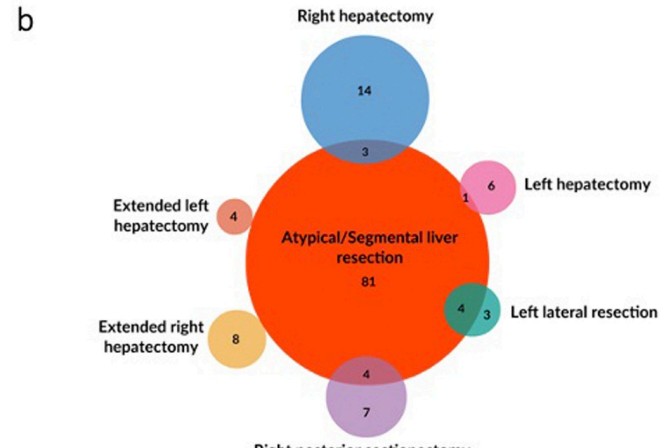

**Fig 2. Trial design.** (a) CONSORT diagram indicating enrolment in the HepaT1ca trial. Median and [IQR] or number and (%). (b) Venn diagram indicating the range of surgical procedures carried out.

## Pre-operative multiparametric MRI correlates with liver histology

Patients displayed a range of tumour and parenchymal liver cT1 values, six examples are shown in Fig 3a–3f. The case shown in Fig 3c depicts a parenchymal cT1 within the normal range [34], and histological analysis of liver tissue adjacent to the tumour explant (white box, Fig 3g) appears typical of a healthy liver showing no inflammation, ballooning or steatosis. The case shown in Fig 3f shows a parenchymal cT1 above the normal healthy range and the associated histology (Fig 3h) indicates ballooned hepatocytes, infiltration of inflammatory cells and grade 3 steatosis.

cT1 correlated with histopathological scoring of ballooning (Spearman's rho = 0.279, p = 0.001) and inflammation (Spearman's rho = 0.276, p<0.001) and PDFF correlated with

**Table 1. Clinicopathologic data for patients enrolled on the HepaT1ca trial prior to treatment.**

| Patient data | n = 143 |
|---|---|
| Age, y, median [IQR] | 64 [56–72] |
| Sex ratio (male to female) | 97:46 |
| Diagnosis, n (%) | |
| Liver metastasis (colorectal primary) | 114 (79.7) |
| Liver metastasis (other primary) | 22 (15.3) |
| Hepatocellular carcinoma | 6 (4.2) |
| Cholangiocarcinoma | 1 (0.6) |
| Known liver disease, n (%) | |
| Alcohol-related | 1 (0.6) |
| NAFLD | 4 (2.8) |
| Hepatitis B | 0 (0) |
| Hepatitis C | 0 (0) |
| Primary biliary cholangitis | 2 (1.4) |
| Primary sclerosing cholangitis | 0 (0) |
| Haemochromatosis | 0 (0) |
| Autoimmune hepatitis | 2 (1.4) |
| Cirrhosis | 3 (2.1) |
| Chemotherapy regime ceased prior to surgery, n (%) | |
| CAPOX | 15 (10.5) |
| FOLFOX | 21 (14.7) |
| FOLFIRI | 9 (6.3) |
| Capecitabine | 7 (4.9) |
| FOLFIRI-cetuximab | 2 (1.4) |
| Other | 17 (11.9) |
| Pre-operative LFT, median [IQR] | |
| Bilirubin (µmol/L) | 10 [8–19] |
| Albumin (µmol/L) | 39 [36–40] |
| AST (U/L) | 26 [22–32] |
| ALT (U/L) | 23 [18–35] |
| ALP (U/L) | 92 [74–119] |
| GGT (U/L) | 39 [26–73] |
| Pre-operative imaging derived metrics, median [IQR] | |
| Planned FLR (%) | 82.8 [61.0–94.6] |
| corrected T1 (ms) | 722.5 [689.2–772.5] |
| PDFF (%) | 3.9 [2.5–8.0] |

steatosis (Spearman's rho = 0.702, P<0.0001). cT1 was significantly elevated in patients with a ballooning score of 2 (* P = 0.0057) or an inflammation score of 2 (*P = 0.011). PDFF showed significant stepwise increases with increasing steatosis scores (*** P<0.001) (Fig 3i–3k).

## Pre-operative multiparametric MRI correlates with length of stay

In the subset of patients where at least 10% of the liver volume was removed (n = 77), the median length of stay for patients with pre-operative cT1 above the upper limit of normal (≥795ms) was 1.5 days longer than those presenting with cT1 within the normal range (Fig 4). Other variables (presence of comorbidities, resection volume greater than 50%, preoperative bilirubin level above normal (20 µmol/L), total lesion volume above 100ml, age over 60 years,

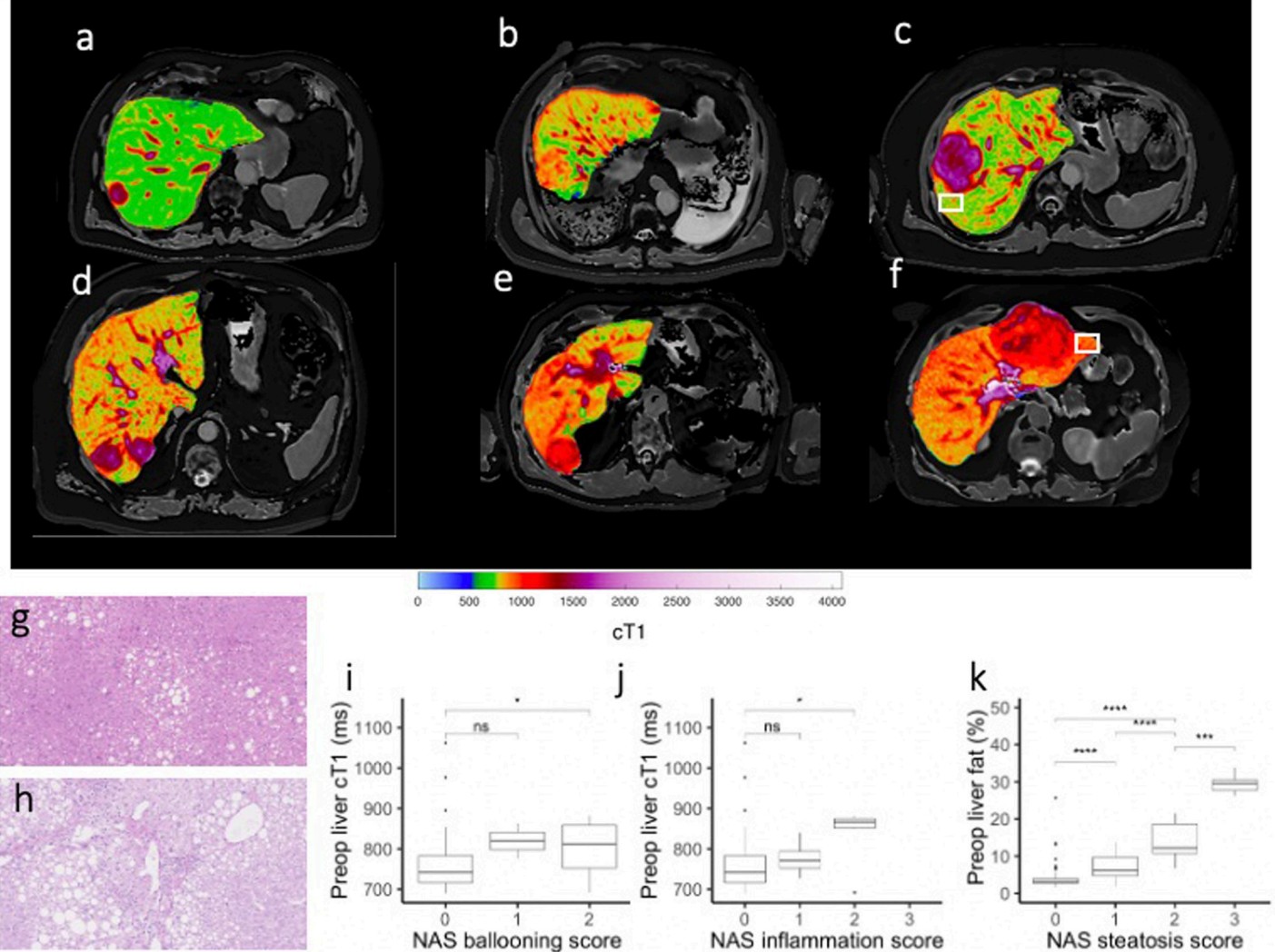

**Fig 3.** (a-f) Quantitative MR images of the liver from six patients presenting with liver cancer using Liver*MultiScan* showing range of cT1 values found in the parenchyma. (g,h) Haematoxylin & eosin stains of parenchymal tissue taken from explant of patient with colorectal liver metastasis (c, white box) showing healthy liver tissue and a patient with hepatocellular carcinoma (f, white box) showing hepatocellular ballooning, clusters of inflammatory cells and regions of steatosis. (I, j, k) Box and Whisker plots of preoperative cT1 or proton density fat fraction (PDFF) against histopathological scoring of ballooning, inflammation or steatosis in explant tissue (1-way ANOVA with Tukey's post-hoc test).

BMI over 25 kg/m2, liver fat over 10%) all showed no significant difference (P>0.05) in length of stay between groups.

## Pre-operative multiparametric MRI predicts with post-operative morbidity

The primary endpoint of this study was to determine the ability of Liver*MultiScan* to predict risk of post-operative morbidity and mortality by measuring the correlation between the pre-operative liver health assessment score and the post-operative liver function composite integer-based risk (Hyder-Pawlik) score; the modified Hyder-Pawlik score (Fig 5a) was also evaluated. Patients with high cT1 had a slightly elevated Hyder-Pawlik score at day 3 than patients with a normal cT1 (P = 0.11 no significant difference), and had a significantly higher modified Hyder-Pawlik score (Fig 5b) than patients with a normal cT1 (P = 0.0076).

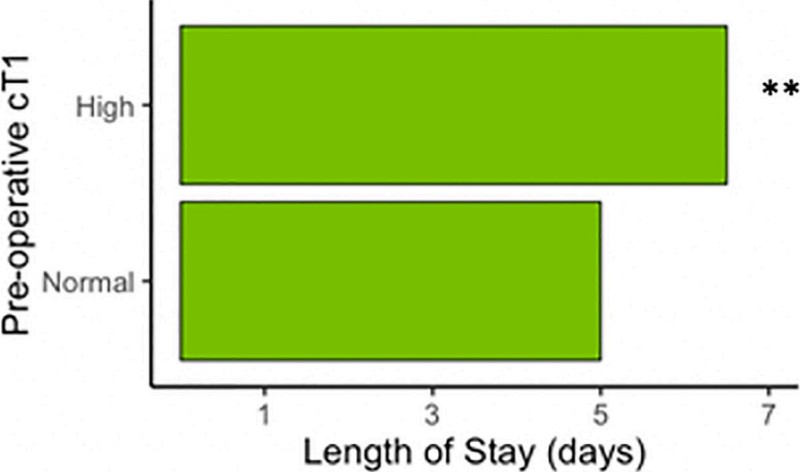

**Fig 4. Length of stay.** Length of stay is 1.5 days longer if there is liver fibroinflammation measured by cT1 in patients with a predicted FLR <90% (high cT1 >795ms; Wilcoxon rank sum test **P = 0.0053, n = 77).

Stepwise logistic regression was performed on the following nine pre-operative variables to determine the diagnostic accuracy to discriminate the patients with a 5-day sum of modified Hyder-Pawlik scores in the upper quartile: age, BMI, creatinine, bilirubin, liver volume, lesion volume, FLR, liver fat and liver cT1. The model was trained using a subset of data (n = 62) in whom at least 10% of the liver volume was removed to exclude patients with very small resections and for whom complete data were available. Non-collinearity of predictors was confirmed using linear regression models. A combination of FLR and pre-operative cT1 had the best performance with an AUROC of 0.78 (95% confidence intervals: 0.66, 0.90) (Fig 5c). The resulting composite biomarker was termed the 'Hepatica score', with a lower FLR and a higher pre-operative cT1 predictive of an increased risk of poor post-operative outcome. This composite biomarker showed a significantly higher diagnostic performance (DeLong's test for two correlated ROC curves P = 0.000104) than FLR alone with an AUROC of 0.70 (95% CI 0.55–0.84).

$$Hepatica\ score = e^{-(1.24+(logFLR*\ -2.26)+(cT1*\ 0.013))}$$

### Hepatica predicts liver regeneration

96 patients returned for a post-operative follow-up mpMRI scan at a mean of 102 ±4.7 days after their operation. The volume of the liver at this timepoint was measured semi-automatically as described above, with 42% of patients regenerating at least 90% of the resected liver volume. The Hepatica score, measured pre-operatively, correlated with the achieved regeneration (Fig 5d, $R^2$ = 0.48, P<0.0001). One example of liver regeneration is depicted; the pre-operative mpMRI indicated a lesion on the right lobe of the liver (Fig 5e), with the resection plan and FLR show in red (Fig 5f). The follow-up scan reveals left lobe hypertrophy (Fig 5g) with a measured increase in total liver volume of 798mL (Fig 5h).

### Discussion

This study aimed to augment current methods for estimating FLP by incorporating non-invasive, contrast-agent free quantitative mpMRI to inform the risk assessment for patients indicated for liver surgery for cancer in the liver. We have shown here that patients with pre-

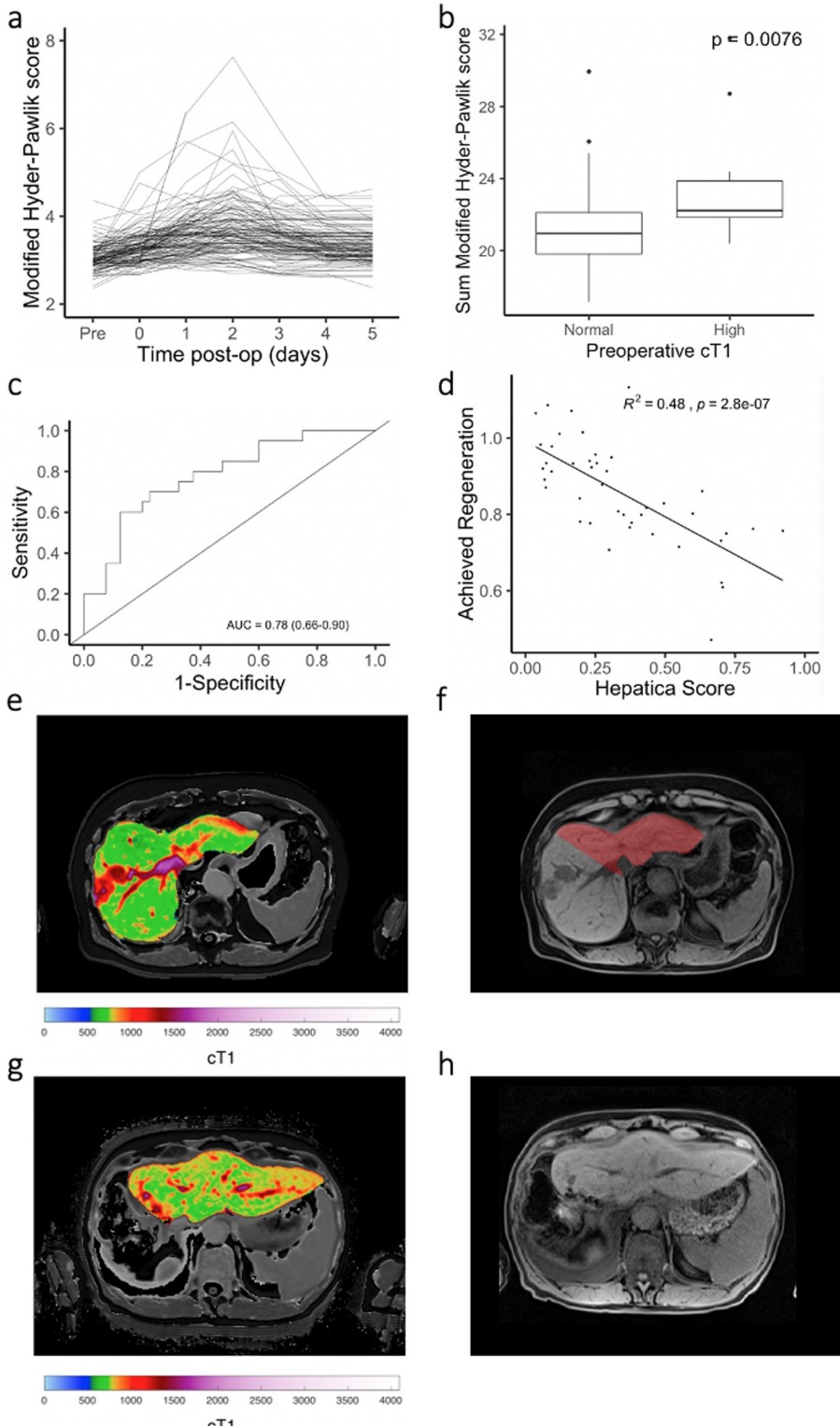

**Fig 5. Post-surgery outcomes.** (a) The modified Hyder-Pawlik score measured before and at each day following surgery (2.5*INR + 17.1*bilirubin + 88.4*creatinine). (b) Sum of the modified Hyder-Pawlik score over 5 days is significantly higher in patients presenting with high preoperative liver cT1 (P = 0.0076). (c) Receiver operating characteristic curve describing the diagnostic ability of the Hepatica score to discriminate the upper quartile of patients representing those with a poor post-operative outcome. (d) Achieved regeneration score (calculated by dividing pre-operative liver volume by the liver volume measured a median of 99 days following resection for liver cancer)

correlated with the pre-operative Hepatica score ($R^2$ = 0.48). (e, f) Exemplar pre-surgical cT1 map and T1-weighted image with FLR overlaid (red) of a patient presenting with a HCC lesion in the right lobe of the liver. (g, h) Post-surgical cT1 map and T1 weighted image of the same patient 3 months following surgery.

operative liver cT1, a biomarker of fibroinflammation, above the upper limit of normal spent on average 1.5 days longer in hospital than those with cT1 in the normal range. Factors affecting length of stay have been investigated previously for liver resection [36, 37] but this is the first report of a non-invasive imaging approach identifying a patient group with a higher length of stay. This is of particular importance given the significant healthcare costs associated with in-patient care (~£10,000 total post-operative costs [38]). The correlations observed for cT1 and PDFF with histological measures of ballooning, inflammation and steatosis in this surgical population, reinforce previously published associations in patients with chronic liver disease [19, 21]. Taken together, this evidence highlights a role for mpMRI in characterising liver disease in place of an invasive liver biopsy, which is typically only performed in selected patients with chronic liver disease as part of the pre-operative work-up, owing to the risk of haemorrhage and tumour seeding [39, 40].

The primary endpoint in this study was to determine the correlation between a composite of FLR and pre-operative liver cT1, with a measure of post-operative morbidity, the Hyder-Pawlik score [4]. The weak correlation seen in this study (multiple R-squared = 0.14) is likely due to low rate of substantial morbidity observed in comparison to that reported by Hyder and colleagues [4]. As a result, our study was insufficiently powered to meet the primary endpoint. The low rate of morbidity is likely due to a combination of factors including liver cancer indication; 18.2% of cases were reported to present with HCC by Hyder and colleagues, while only 6 of 143 (4.2%) cases in our study presented with HCC. Additionally, 9.1% of cases investigated by Hyder and colleagues presented with underlying liver cirrhosis whereas no patients in our study were found to have histological cirrhosis (modified ISHAK fibrosis score 6).

The Hyder-Pawlik score identifies an increased risk of 90-day mortality when measured as >9 points at day 3 post-surgery. Prediction of 90-day mortality using patient information collected after the surgical procedure has utility in enabling clinical decisions regarding the level of post-operative care required for the patient. However, additional patient outcome and management benefits may be gained if such analysis was carried out pre-operatively. In this study, we opted to investigate imaging-based pre-surgical variables that would permit the evaluation of the risk of the surgical procedure before it was carried out. The composite biomarker 'Hepatica score' generated in our study performed well at classifying the quartile of patients with poor liver function as measured on each of the five days following surgery (modified Hyder-Pawlik score). We propose that this approach better captures the transient changes in serum liver enzymes that are indicative of limited liver function [41] and are used in clinical monitoring to guide changes in patient management during post-operative recovery.

Another key element in the recovery of patients following liver resection is the capacity of the liver to regenerate. In this study, the pre-operative Hepatica score showed excellent correlation with the volume of liver regenerated at the 3-month post-operative MRI scan. This highlights the utility of augmenting FLR with a measure of liver health, in this case the fibroinflammatory biomarker, cT1. Our understanding of the processes regulating liver regeneration are evolving, but inflammation [42] and recent chemotherapy prior to resection [43] have previously been reported as important modifiers.

A limitation of this study is the relatively small number of patients with poor post-operative liver performance. In order to build suitably robust and transferable predictive models, a sufficiently wide distribution of patients is required. As this study was prespecified to enrol all-

comers at tertiary referral centres in the UK, the patients recruited reflected the relatively low prevalence of primary liver cancer in this region with the majority having metastatic liver cancer on a background of relatively healthy liver tissue. A second limitation is our use of a new measure of post-operative liver performance, the modified Hyder-Pawlik score. This approach was derived to provide increased sensitivity to more subtle derangements in post-operative liver function and to allow an understanding of the acute care requirements and level of observation required for patients still under the care of the surgeon and associated multidisciplinary team. This scoring system and its clinical validity requires further evaluation in future studies.

In summary, the results presented in this study highlight the utility of pre-operative mpMRI for predicting post-operative liver performance and associated length of stay in hospital. This has the potential to transform surgical decision-making and improve personalised risk assessment for patients undergoing liver resection for cancer. Further evaluation of this technology is needed to fully evaluate the clinical utility in a broader patient population.

## Supporting information

**S1 Fig. (a) Distribution of Hyder-Pawlik scores measured 3 days following surgery.**
(TIF)

## Acknowledgments

We are extremely grateful for the assistance of the following individuals and teams: Edinburgh Imaging QMRI Facility; NHS Lothian Clinical Research Facility team; Paul Fitch, Olga Garcia Sabater, Lyndsey Woolrych, Katie Andreucetti, Sheila Marshall, Susan Bodie, Murray Britton, Susan Keggie.

HepaT1ca Study Group: elicia Bachtiar[1], Henry Willman[1], Anya Adair[2], Ewan Brown[2], O James Garden[2], Jim Gordon-Smith[2], Ewen Harrison[2], Andrew Healey[2], Hamish Ireland[2], Neil Masson[2], Rowan Parks[2], James Powell[2], Ravi Ravindran[2], Sarah Thomasset[2], Stephen J Wigmore[2]; Ben Cresswell[3], Timothy John[3], Asmat Mustajab[3], Delia Peppercorn[3], Karen Scott[3], Andrew Thrower[3].

Velicia.bachtiar@perspectum.com

1. Perspectum, Gemini One, 5520 John Smith Drive, Oxford, UK
2. University of Edinburgh, Edinburgh, UK
3. Hampshire Hospitals Foundation Trust, Basingstoke, UK

## Author Contributions

**Conceptualization:** Damian J. Mole, Jonathan A. Fallowfield, Timothy Kendall, Matt Kelly, Gerard Ridgway, John J. Connell, Rajarshi Banerjee, J. Michael Brady, Joanne McClintock, Fenella Welsh, Myrddin Rees.

**Data curation:** John McGonigle, Anthony Wackett.

**Formal analysis:** Ahmed E. Sherif, Timothy Kendall, Matt Kelly, Gerard Ridgway, John J. Connell, Myrddin Rees.

**Funding acquisition:** Damian J. Mole, Matt Kelly, Rajarshi Banerjee, J. Michael Brady, Joanne McClintock, Fenella Welsh, Myrddin Rees.

**Investigation:** Damian J. Mole, Jonathan A. Fallowfield, Ahmed E. Sherif, Timothy Kendall, Scott Semple, Xiaozhong Zheng, Michael Hughes, Lucile Neyton, Joanne McClintock, Garry Tucker, Hilary Nailon, Dilip Patel, Michelle Steven, Fenella Welsh, Myrddin Rees.

**Methodology:** Timothy Kendall, Scott Semple, Matt Kelly, Gerard Ridgway, John J. Connell, John McGonigle, Rajarshi Banerjee, J. Michael Brady, Fenella Welsh, Myrddin Rees.

**Software:** Gerard Ridgway, John McGonigle.

**Supervision:** J. Michael Brady.

**Visualization:** Gerard Ridgway, John J. Connell, John McGonigle.

**Writing – original draft:** Damian J. Mole, John J. Connell.

**Writing – review & editing:** Damian J. Mole, Jonathan A. Fallowfield, Timothy Kendall, Matt Kelly, Gerard Ridgway, John J. Connell, John McGonigle, Rajarshi Banerjee, J. Michael Brady, Joanne McClintock, Fenella Welsh, Myrddin Rees.

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
