## [Decision Letter · Decision Letter 0]

22 Jul 2020

PONE-D-20-17935

Quantitative magnetic resonance imaging predicts individual future liver performance after liver resection for cancer.

PLOS ONE

Dear Dr. Connell,

Thank you for submitting your manuscript to PLOS ONE. After careful consideration, we feel that it has merit but does not fully meet PLOS ONE’s publication criteria as it currently stands. Therefore, we invite you to submit a revised version of the manuscript that addresses the points raised during the review process.

We look forward to receiving your revised manuscript.

Kind regards,

Leonidas G Koniaris, MD

Academic Editor

PLOS ONE

Additional Editor Comments:

Please address review comments.

2. Please include your table as part of your main manuscript and remove the individual file.

Please note that supplementary tables should be uploaded as separate "supporting information" files.

4. Thank you for stating the following in the Financial Disclosure section:

'We gratefully acknowledge the funders of this work through an Early Stage Biomedical Catalyst Award from Innovate UK, part of UK Research and Innovation, ref: 79891-510341. DJM is funded by the UK Medical Research Council through a Senior Clinical Fellowship, ref: MR/P008887/1. JAF was funded through an NHS Research Scotland/Universities Senior Clinical Fellowship. Additional funding was contributed by North Hampshire Medical Fund.'

We note that one or more of the authors are employed by a commercial company: Perspectum

5. One of the noted authors is a group; HepaT1ca Study Group.

In addition to naming the author group, please list the individual authors and affiliations within this group in the acknowledgments section of your manuscript.

Please also indicate clearly a lead author for this group along with a contact email address.

6. Please include captions for your Supporting Information files at the end of your manuscript, and update any in-text citations to match accordingly. Please see our Supporting Information guidelines for more information: http://journals.plos.org/plosone/s/supporting-information

Reviewers' comments:

Reviewer's Responses to Questions

**Comments to the Author**

1. Is the manuscript technically sound, and do the data support the conclusions?

Reviewer #1: Partly

2. Has the statistical analysis been performed appropriately and rigorously? 

Reviewer #1: Yes

3. Have the authors made all data underlying the findings in their manuscript fully available?

Reviewer #1: Yes

4. Is the manuscript presented in an intelligible fashion and written in standard English?

Reviewer #1: Yes

5. Review Comments to the Author

Reviewer #1: The authors uses quantitative MRI to determine pre-operative liver health and combined this with future liver remnant measurements to predict future liver performance after liver resection. The results showed that a combination of low future liver remnant volume and elevated corrected T1 liver measurements were able to predict poor future liver performance. While the work shows significant potential there are several critiques and unanswered questions

1. The authors do not define what criteria was used to determine poor post-operative liver performance. Please define this criteria and show the comparison between the high risk group and the low risk group for predictive post-operative poor liver function.

2. Length of stay of 1.5 days long in the high risk group compared to the low risk group. Were other factors that may have contributed to this, such as age, comorbidities, extent of resection, etc., controlled for?

3. The AUROC and confidence intervals for the predictive value of FLR and the hepatica score strongly overlap. I am not sure it can definitively be said the hepatica score is a better predictor. If anything it adds only a modest increase in predictive value.

4. Only 21% of the patients had abnormal liver parenchyma and none had histopathologic proven cirrhosis. These are the patients that a predictive score would be most applicable to. It would be ideal to see a larger percentage of patients with abnormal liver parenchyma and to include patients with cirrhosis in order to draw stronger conclusions that would be more broadly applicable.

6. PLOS authors have the option to publish the peer review history of their article (what does this mean?). If published, this will include your full peer review and any attached files.

Reviewer #1: No

---

## [Author Response · Author response to Decision Letter 0]

3 Aug 2020

All reviewer comments are responded to in attached document.

---

## [Editor Report · Decision Letter 1]

20 Aug 2020

Quantitative magnetic resonance imaging predicts individual future liver performance after liver resection for cancer.

PONE-D-20-17935R1

Dear Dr. Connell,

We’re pleased to inform you that your manuscript has been judged scientifically suitable for publication and will be formally accepted for publication once it meets all outstanding technical requirements.

Kind regards,

Leonidas G Koniaris, MD

Academic Editor

PLOS ONE
---

## [Editor Report · Acceptance letter]

9 Nov 2020

PONE-D-20-17935R1 

Quantitative magnetic resonance imaging predicts individual future liver performance after liver resection for cancer. 

Dear Dr. Connell:

I'm pleased to inform you that your manuscript has been deemed suitable for publication in PLOS ONE. Congratulations! Your manuscript is now with our production department. 

Kind regards, 

on behalf of

Dr. Leonidas G Koniaris 

Academic Editor

PLOS ONE